# The efficacy of inspiratory muscle training in patients with coronary artery disease: Protocol for a systematic review and meta-analysis

**Yoshito Kadoya**[1], **Saad Balamane**[1,2], **Sarah Visintini**[3], **Benjamin Chow**[1]*

**1** Division of Cardiology, University of Ottawa Heart Institute, Ottawa, Ontario, Canada, **2** Faculty of Health Sciences, Queen's University, Kingston, Ontario, Canada, **3** Berkman Library, University of Ottawa Heart Institute, Ottawa, Ontario, Canada

* bchow@ottawaheart.ca

**Data Availability Statement:** Deidentified research data will be made publicly available when the study is completed and published.

## Abstract

### Background

Inspiratory muscle training (IMT) has been recognized as an effective form of training in patients with cardiovascular disease and heart failure. However, little is known about the efficacy of IMT in the treatment of patients with coronary artery disease (CAD). The aim of this systematic review will be to evaluate randomized controlled studies to understand the effect of IMT on CAD patients.

### Method

We will include randomized controlled trials evaluating the efficacy of IMT in patients 18 years and older diagnosed with CAD. Crossover trials, cluster-randomized, quasi-randomized, and non-randomized trials will be excluded. Study search will be conducted in major databases (MEDLINE, the Cochrane Central Register of Controlled Trials, Embase, and PEDro). The study intervention will be IMT independent of the duration, frequency, or intensity of training. The primary outcome will be quality of life, patient-reported health status, and all adverse events related to IMT. Secondary outcomes will include exercise capacity and respiratory muscle strength. The risk of bias will be evaluated based on the Cochrane Risk of Bias tool. Screening, data extraction, and quality assessment will be performed by two independent reviewers. If two or more studies are considered to be clinically homogeneous, a meta-analysis based on the random-effects model will be performed. The quality of evidence will be evaluated based on the GRADE approach.

### Conclusion

This systematic review will improve our understanding of the effects of IMT on CAD patients and potentially establish IMT as an alternative form of exercise training for the treatment of CAD.

**Funding:** The authors received no specific funding for this work.

**Competing interests:** The authors have declared that no competing interests exist.

## Trial registration

**Study registration.** OSF registries (https://osf.io/3ch7m). Date registered: May 10, 2022. Registration DOI: https://doi.org/10.17605/OSF.IO/GVMY7.

## Introduction

Coronary artery disease (CAD) is a leading cause of morbidity and mortality [1]. The health benefits of regular physical activity have been well-established [2], and exercise training is one of the mainstays of CAD prevention [3]. Recent guidelines recommend that adults perform at least 150 minutes/week of moderate-intensity aerobic physical activity or 75 minutes/week of vigorous-intensity aerobic physical activity [4]. However, roughly half of all adults in North America do not achieve this minimum recommended level of physical activity [4,5]. Factors leading to low levels of physical activity may be related to a variety of issues, such as lack of time, access to facilities, and financial costs. Effective and well-tolerated alternative forms of exercise are needed [6].

Inspiratory muscle training (IMT) is a form of exercise that engages the diaphragm and accessory respiratory muscles to repeatedly inhale against resistance [7–9]. A previous meta-analysis showed that IMT can improve pulmonary function, exercise tolerance, and quality of life of patients with chronic heart failure and relieve symptoms of dyspnea [10]. A randomized control study has also shown that high-resistance IMT can lower blood pressure and improve vascular endothelial function [9]. The authors concluded that this effect was in part by increasing nitric oxide bioavailability via increased endothelial nitric oxide synthase activation and reduced reactive oxygen species production and oxidative stress [9]. Based on these results, IMT could have beneficial effects in patients with cardiovascular disease, especially CAD. Improvements in endothelial function of coronary arteries could improve coronary blood flow, leading to the improvement of anginal symptoms as well as quality of life.

Huzmel et al. demonstrated that IMT improved not only functional exercise capacity and pulmonary functions but also health-related quality of life in patients with CAD [10]. However, there is limited data on the benefits of IMT in CAD patients. To the best of our knowledge, there have been no systematic reviews conducted to assess the impact of IMT on CAD. Further research is required to investigate patient-centered outcomes in CAD patients undergoing IMT.

### Objectives

We aim to review the effect of IMT (either alone or in combination with aerobic training) in patients with CAD. The specific clinical questions will be: (1) Can IMT improve quality of life in patients with CAD? and (2) Can IMT improve patient-reported health status in patients with CAD? Our results may identify a novel and effective alternative to conventional aerobic exercise therapy for the treatment of CAD.

### Methods

This protocol is developed by following the preferred reporting items for systematic review and meta-analysis 2015 (PRISMA-P) guidelines [10]. Our results will be reported in adherence to the PRISMA Statement for Reporting Systematic Reviews and Meta-Analyses [11]. The checklist is shown in S1 Appendix. Our protocol for this research has been registered on OSF registries (https://osf.io/3ch7m).

## Information sources

A search will be performed using MEDLINE (Ovid), the Cochrane Central Register of Controlled Trials (Ovid), EMBASE (Ovid), and PEDro (https://pedro.org.au/). We will also search the World Health Organization International Clinical Trials Platform Search Portal (ICTRP) and ClinicalTrials.gov for ongoing or unpublished trials. Additionally, manual searches, such as checking reference lists and searching conference proceedings, will also be added to minimize missing clinical studies.

## Search strategy

Following the PRESS guidelines [12], the search strategy will be developed by our institutional librarian and will also be peer-reviewed by a second librarian.

Search terms associated with IMT and CAD will be included (S2 Appendix).

## Inclusion criteria

**Population.** We will review studies studying adults $\geq$ 18 years diagnosed with CAD (angiographically or clinically symptomatic CAD, irrespective of a history of myocardial infarction, coronary artery bypass graft [CABG], and percutaneous coronary intervention [PCI]). Studies in which more than half of the participants had other non-coronary cardiac comorbidities such as valvular disease, heart failure, or arrhythmia will be excluded unless CAD patients can be extracted. Studies in which IMT is performed combined with revascularization after patient enrollment, e.g., IMT shortly before or after PCI/CABG during the same hospitalization, will be excluded because the effects of IMT and revascularization cannot be evaluated separately.

**Inspiratory muscle training.** IMT consists of performing repeated inspirations against resistance with unopposed (no resistance) expirations [13]. IMT is performed using either a constant resistance or a pressure-threshold device. Studies with mixed interventions (e.g., IMT combined with aerobic exercise or resistance training) will be included. No criteria will be set on the duration, frequency, or intensity of IMT. The control group will include non-exercise usual care, sham-training, any kind of exercise training, and respiratory training with no load or with a lower load than that of intervention group.

**Outcomes.** Primary outcomes will include health-related quality of life assessed using validated instruments (e.g., SF-36, EQ-5D, Minnesota living with heart failure questionnaire), patient-reported health status, such as angina, dyspnea, depression, functional status, assessed using validated instruments (e.g., SAQ-7, PHQ-2, modified Medical Research Council scale), and all adverse events related to IMT.

Secondary outcomes will include exercise capacity (e.g., peak oxygen consumption [peak VO2]) and respiratory muscle strength (e.g., Maximal inspiratory pressure [MIP] and maximal expiratory pressure [MEP]).

**Study design included.** In this review, to collect studies with high level of evidence and to minimize heterogeneity of studies, only randomized controlled trials will be included. Cross-over trials, cluster-randomized, quasi-randomized, and non-randomized trials will be excluded. Non-randomized interventional studies, prospective or retrospective cohort studies, case-control studies, letters, editorials, review articles, and case reports will be also excluded.

## Eligibility criteria

We will include all papers including published, unpublished articles, conference abstracts, and letters. We will not apply language or country restrictions. We will not exclude studies based

on the observation period or publication year. If studies reported mixed populations, authors will be contacted to obtain stratified data for patients with CAD, but if stratified data cannot be made available, studies will be excluded.

## Selection process

Titles and abstracts will be screened independently by two reviewers. Thereafter, the assessment of the eligibility based on the full texts will be reviewed independently by two reviewers. Discrepancies between the two reviewers will be resolved by discussion, and if a resolution cannot be achieved, a third reviewer will act as an arbiter. Original authors will be contacted if relevant data is missing. The rationale for excluding full texts will be recorded. If there are two or more papers that reported the same outcome in the same study, only the study containing the larger participation will be selected. The selection of eligible articles will be summarized in a PRISMA flowchart.

## Data collection process

Data extraction of the included studies will be performed by two independent reviewers using a pre-designed, standardized data extraction sheet. The form will include information on study design, study population, interventions (including the intensity, duration, and frequency of IMT), and outcomes (patient-reported health status and health-related quality of life). For calibration, reviewers will compare the extraction sheet on two randomly selected studies. Once calibrated, they will independently collect the predefined information. Similar to above, disagreements will be resolved by consensus, and if needed, a third reviewer will act as an arbiter. If necessary, study authors will be contacted by email for additional information and when needed, follow-up emails will be sent at 2 and 4 weeks.

## Data items

For each study, publication information (e.g., authors' name, year of publication, publication country), study information (e.g., number of participants, inclusion/exclusion criteria, funding resource), participants' characteristics (age, sex, comorbidity, medication, history of PCI/CABG), and information on IMT (e.g., type of device, duration, frequency, intensity). The pre-defined outcomes will be extracted according to IMT and control for each group in each study.

## Risk of bias in individual studies

Risk of bias will be evaluated based on the Cochrane Risk of Bias tool (RoB 2) [14]. If there is missing data, we will consider the risk of bias as 'unclear' and try to contact the authors for additional information. Two reviewers will evaluate the risk of bias independently, and any disagreements will be resolved through discussion until consensus is reached, or with the assistance of a third reviewer.

## Data synthesis

The clinical heterogeneity of the studies will be assessed based on the characteristics of the study participants, study design, and the methods used to evaluate the outcomes. In cases where two or more studies are deemed clinically homogeneous, a meta-analysis will be conducted using a random effects model.

For the primary outcome, the mean differences and 95% confidence intervals (CIs) will be pooled for the following continuous variables: health-related quality of life, patient-reported

health status, and the incidence rate of adverse events. Given that the measures of quality of life are likely to vary across studies, a standardized change score (standardized mean difference, SMD) will be calculated for each study. The mean differences and 95% CIs will be pooled for the secondary outcome, including indices of exercise capacity and respiratory muscle strength. If there is insufficient data to conduct a meta-analysis, descriptive results will be provided.

The statistical heterogeneity will be evaluated by visually inspecting the forest plots and calculating the I2 statistic. In instances of substantial heterogeneity (I2>50%), we plan to identify the sources of heterogeneity through subgroup and sensitivity analyses. The Cochrane Chi2 test (Q-test) will be performed for the I2 statistic, and a P value less than 0.10 will be considered statistically significant. Finally, the potential publication bias will be assessed through visual inspection of the funnel plot and/or Trim and fill method. To address publication bias, we may employ trim and fill or Egger's linear regression test in post-hoc analyses. All statistical analyses will be conducted utilizing Review Manager 5.1.

## Confidence in cumulative evidence

The quality of evidence will be evaluated based on the GRADE (Grading of Recommendations Assessment, Development and Evaluation) approach [15]. The five GRADE considerations (i.e., risk of bias, consistency of effect, imprecision, indirectness, and publication bias) will be used to draw conclusions about the certainty of evidence by two independent reviewers.

## Patient and public involvement statement

This literature review will not require direct patient involvement.

## Discussion

IMT has been demonstrated as a safe and practical exercise program for a diverse range of diseases through multiple studies [16–23]. Its efficacy in treating cardiac diseases such as heart failure and hypertension has been widely acknowledged, yet there is limited evidence available to support its potential application in CAD patients [9,21–23].

Physical exercise is widely accepted as crucial for the secondary prevention of CAD patients, however, adherence to conventional exercise programs remains low. On the other hand, a study by Craighead et al. showed that 95% of patients adhered to 30 minutes of IMT per week [9]. IMT has also been shown to be well tolerated and safe in several studies [16,17,21–23].

This systematic review aims to gain a better understanding of the potential of IMT as a non-pharmacological intervention in CAD patients by evaluating its efficacy in managing this patient population.

Our review has several advantages, including a comprehensive literature search that encompasses all published and unpublished sources such as conference abstracts, letters, and references to previous reviews, without restrictions on language or country. Furthermore, the screening, extraction, and quality assessment steps will be performed independently by two separate reviewers. The GRADE tool will be utilized to assess the quality of the evidence from the retrieved studies. To the best of our knowledge, this review is the first to evaluate the clinical utility of IMT for patient-centered outcomes in CAD patients.

However, there are also several limitations to consider. Despite the numerous studies reporting the clinical benefits of IMT, there are no established training protocols. This may result in heterogeneity in training intensity, frequency, duration, and devices used in each study. We will address this limitation through subgroup analysis. Furthermore, pooling the

results of different IMT studies and protocols may increase heterogeneity. Additionally, evaluating patient-centered outcomes such as symptoms and quality of life can pose challenges, as they are subjective and may be evaluated through multiple methodologies and scales. In such cases, only one indicator will be selected for meta-analysis, which may introduce selection bias. Subgroup analyses will also be conducted if possible.

Given the aging population, increasing frailty, and declining physical activity levels, it is imperative to validate training methods that are widely applicable, feasible, and time-efficient for CAD patients. Our review aims to provide insights into the potential of IMT as a non-pharmacological intervention for this patient population.

## Conclusion

In conclusion, this systematic review aims to evaluate the evidence for the effectiveness of IMT in improving health-related quality of life and patient-reported health status in patients with CAD. A comprehensive search strategy and strict inclusion and exclusion criteria will be used to identify relevant studies. The results of this review will provide a synthesis of the existing evidence on IMT for CAD and inform future research in this area. It is expected that the systematic review will contribute to the development of novel training approaches and establish IMT as a potential treatment option for CAD.

## Supporting information

**S1 Appendix. PRISMA-P 2015 checklist.**
(DOCX)

**S2 Appendix. Search strategy.**
(DOCX)

## Author Contributions

**Conceptualization:** Yoshito Kadoya.

**Investigation:** Sarah Visintini.

**Methodology:** Yoshito Kadoya, Saad Balamane, Sarah Visintini, Benjamin Chow.

**Supervision:** Benjamin Chow.

**Validation:** Saad Balamane.

**Writing – original draft:** Yoshito Kadoya.

**Writing – review & editing:** Benjamin Chow.

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
