## [Decision Letter · Decision Letter 0]

25 Jan 2023

PONE-D-22-16997The efficacy of inspiratory muscle training in patients with coronary artery disease: Protocol for a systematic review and meta-analysisPLOS ONE

Dear Dr. Chow,

Thank you for submitting your manuscript to PLOS ONE. After careful consideration, we feel that it has merit but does not fully meet PLOS ONE’s publication criteria as it currently stands. Therefore, we invite you to submit a revised version of the manuscript that addresses the points raised during the review process.

We look forward to receiving your revised manuscript.

Kind regards,

Charlotte Beaudart

Academic Editor

PLOS ONE

https://journals.plos.org/plosone/s/fileid=ba62/PLOSOne_formatting_sample_title_authors_affiliations.pdf.

2. Thank you for submitting the above manuscript to PLOS ONE. During our internal evaluation of the manuscript, we found significant text overlap between your submission and previous work in the methods and discussion. We would like to make you aware that copying extracts from previous publications, especially outside the methods section, word-for-word is unacceptable. In addition, the reproduction of text from published reports has implications for the copyright that may apply to the publications.

Please revise the manuscript to rephrase the duplicated text, cite your sources, and provide details as to how the current manuscript advances on previous work. Please note that further consideration is dependent on the submission of a manuscript that addresses these concerns about the overlap in text with published work.

We will carefully review your manuscript upon resubmission and further consideration of the manuscript is dependent on the text overlap being addressed in full. Please ensure that your revision is thorough as failure to address the concerns to our satisfaction may result in your submission not being considered further.

3. PLOS requires an ORCID iD for the corresponding author in Editorial Manager on papers submitted after December 6th, 2016. Please ensure that you have an ORCID iD and that it is validated in Editorial Manager. To do this, go to ‘Update my Information’ (in the upper left-hand corner of the main menu), and click on the Fetch/Validate link next to the ORCID field. This will take you to the ORCID site and allow you to create a new iD or authenticate a pre-existing iD in Editorial Manager. Please see the following video for instructions on linking an ORCID iD to your Editorial Manager account: https://www.youtube.com/watch?v=_xcclfuvtxQ.

Additional Editor Comments :

Please update the abstract using the PRISMA-A guidelines.

Objective: I have a confusion about the primary outcome. Is it quality of life as mentioned in the abstract or anginal symptoms as mentioned in the objective ? Please be more specific in the objective about the primary and secondary outcomes and make sure to be consistent with other parts of the protocol.

Methods: Should be restructured. Please start with information source and search strategy. Then, provide inclusion criteria with the PICO and then eligibility criteria. Study designs included could be part of the inclusion criteria (S from the PICOS formulation). Please also justify why only RCT will be included (see AMSTAR critical appraisal tool criteria).

Information source: I recommend authors to add other types of manual search such as searching in clinical trials. Please read AMSTAR2 critical appraisal tool to be sure to meet the proposed criteria.

Search strategy: please provide the full search strategy in appendix. Providing only keywords that will be used to develop the search strategy is not enough.

Statistical analyses: please already defined the subgroups analyses as well as the sensitivity analyses that will be performed. Which software will be used for analyses ? Why only a visual inspection of the funnel plot will be performed for the assessment of publication bias? Statistical tests already exists. Also, do the authors plan to do something is publication bias is present? Once again, I also have a confusion about the outcomes investigated (lack of consistency between abstract, objectives and statistical part).

Reviewers' comments:

Reviewer's Responses to Questions

**Comments to the Author**

1. Does the manuscript provide a valid rationale for the proposed study, with clearly identified and justified research questions?

Reviewer #1: Partly

Reviewer #2: Yes

2. Is the protocol technically sound and planned in a manner that will lead to a meaningful outcome and allow testing the stated hypotheses?

Reviewer #1: Yes

Reviewer #2: Yes

3. Is the methodology feasible and described in sufficient detail to allow the work to be replicable?

Reviewer #1: Yes

Reviewer #2: Yes

4. Have the authors described where all data underlying the findings will be made available when the study is complete?

Reviewer #1: Yes

Reviewer #2: Yes

5. Is the manuscript presented in an intelligible fashion and written in standard English?

Reviewer #1: Yes

Reviewer #2: Yes

6. Review Comments to the Author

You may also provide optional suggestions and comments to authors that they might find helpful in planning their study.

Reviewer #1: Dear Authors

The recommendations for study protocol of the systematic review: The efficacy of inspiratory muscle training in patients with coronary artery disease: Protocol for a systematic review and meta-analysis, PONE-D-22-16997

I believe that first, the authors can expose in the pilot search strategy which studies in the main databases answer the research question to understand the effect of IMT on anginal symptoms in coronary artery disease.

In the introduction, on line 62; refers to just a single RCT demonstrating the effect of IMT in improving vascular endothelial function and oxide bioavailability in normal adults.

Perhaps a more comprehensive search strategy can provide more data to justify that there are studies in the literature that support the proposed systematic review.

The systematic review protocol record can include in addition to the record number; the day, month and year.

Despite line 218 is reporting that the search will be comprehensive, on line 117, with information sources, is not include other databases such as; CINAHL (Cumulative Index to Nursing and allied Health, Web of Science)

The conclusion does not fully answer the research question in question.

Reviewer #2: The protocol is well written and, in my opinion, meets the criteria for publication, as this is an option of PLOS ONE.

7. PLOS authors have the option to publish the peer review history of their article (what does this mean?). If published, this will include your full peer review and any attached files.

Reviewer #1: No

Reviewer #2: **Yes: **Giulliano Gardenghi

---

## [Author Response · Author response to Decision Letter 0]

1 May 2023

Manuscript ID: PONE-D-22-16997

First, we would like to thank the editor and the reviewers for their careful review and useful suggestions. We truly appreciate their informative advice. The comments were extremely valuable and have helped us to revise and improve our manuscript. We have responded to all comments from the editor and reviewers below. We hope this revision process has addressed all concerns related to the original manuscript. Changes and corrections are indicated using red font in the revised manuscript.

Editor Comments:

https://journals.plos.org/plosone/s/fileid=ba62/PLOSOne_formatting_sample_title_authors_affiliations.pdf

Response:

We have referred to the manuscript body formatting guidelines and revised our manuscript based on the guideline. 

2. Thank you for submitting the above manuscript to PLOS ONE. During our internal evaluation of the manuscript, we found significant text overlap between your submission and previous work in the methods and discussion. We would like to make you aware that copying extracts from previous publications, especially outside the methods section, word-for-word is unacceptable. In addition, the reproduction of text from published reports has implications for the copyright that may apply to the publications.

Please revise the manuscript to rephrase the duplicated text, cite your sources, and provide details as to how the current manuscript advances on previous work. Please note that further consideration is dependent on the submission of a manuscript that addresses these concerns about the overlap in text with published work.

We will carefully review your manuscript upon resubmission and further consideration of the manuscript is dependent on the text overlap being addressed in full. Please ensure that your revision is thorough as failure to address the concerns to our satisfaction may result in your submission not being considered further.

Response:

Thank you for bringing the text overlap issue to our attention. We understand the importance of properly citing sources and avoiding plagiarism in scientific publication.

We apologize for any oversight in our initial submission and have revised the manuscript to address the concerns raised. We rephrased the duplicated text and properly cited our sources in accordance with standard academic practices.

In the Discussion and Conclusion sections, although the argument has not been changed, the English wording has been changed throughout, and we appreciate your understanding.

Lines 358-466

3. PLOS requires an ORCID iD for the corresponding author in Editorial Manager on papers submitted after December 6th, 2016. Please ensure that you have an ORCID iD and that it is validated in Editorial Manager. To do this, go to ‘Update my Information’ (in the upper left-hand corner of the main menu), and click on the Fetch/Validate link next to the ORCID field. This will take you to the ORCID site and allow you to create a new iD or authenticate a pre-existing iD in Editorial Manager. Please see the following video for instructions on linking an ORCID iD to your Editorial Manager account: https://www.youtube.com/watch?v=_xcclfuvtxQ.

Response:

We have uploaded ORCID iD for the corresponding author as requested.

Additional Editor Comments:

1. Please update the abstract using the PRISMA-A guidelines.

Response:

As suggested, we have revised the abstract based on the PRISMA-A guidelines.

2. Objective: I have a confusion about the primary outcome. Is it quality of life as mentioned in the abstract or anginal symptoms as mentioned in the objective? Please be more specific in the objective about the primary and secondary outcomes and make sure to be consistent with other parts of the protocol.

Response:

The aim of this systematic review will be to understand the effect of IMT on both quality of life and anginal symptoms in CAD patients. The specific clinical questions will be: (1) Can IMT improve quality of life in patients with CAD? and (2) Can IMT improve patient-reported health status in patients with CAD?

Primary outcome in our review will be quality of life, patient-reported health status, and all adverse events related to IMT.

To clarify this, we modified the Introduction and Objective sections as follows.

Lines 90-97 

Huzmel et al. demonstrated that IMT improved not only functional exercise capacity and pulmonary functions but also health-related quality of life in patients with CAD [10]. However, there is limited data on the benefits of IMT in CAD patients. To the best of our knowledge, there have been no systematic reviews conducted to assess the impact of IMT on CAD. Further research is required to investigate patient-centered outcomes in CAD patients undergoing IMT.

Lines 99-102

We aim to review the effect of IMT (either alone or in combination with aerobic training) in patients with CAD. The specific clinical questions will be: (1) Can IMT improve quality of life in patients with CAD? and (2) Can IMT improve patient-reported health status in patients with CAD? Our results may identify a novel and effective alternative to conventional aerobic exercise therapy for the treatment of CAD.

3. Methods: Should be restructured. Please start with information source and search strategy. Then, provide inclusion criteria with the PICO and then eligibility criteria. Study designs included could be part of the inclusion criteria (S from the PICOS formulation). Please also justify why only RCT will be included (see AMSTAR critical appraisal tool criteria).

Response:

According to your suggestion, we have restructured the Methods section. 

Methods section includes 1) Inclusion criteria (PICO), 2) Study design included, 3) Eligibility criteria, 4) Selection process, 5) Data collection process, 6) Data items, 7) Risk of bias in individual studies, 8) Data synthesis, 9) Confidence in cumulative evidence, 10) Patient and public involvement statement.

Lines 112-178

4. Information source: I recommend authors to add other types of manual search such as searching in clinical trials. Please read AMSTAR2 critical appraisal tool to be sure to meet the proposed criteria.

Response:

As suggested, we will add manual searches to minimize missing clinical studies.

We have added the following sentence in the manuscript.

Line 125-127

Additionally, manual searches, such as checking reference list and searching conference proceedings, will also be added to minimize missing clinical studies.

5. Search strategy: please provide the full search strategy in appendix. Providing only keywords that will be used to develop the search strategy is not enough.

Response:

We did have the search strategies provided in the document entitled "S2 Appendix". Is it possible that the reviewer was referring to full result counts and dates run? If so, we have provided a more detailed version for review.

6. Statistical analyses: please already defined the subgroups analyses as well as the sensitivity analyses that will be performed. Which software will be used for analyses? Why only a visual inspection of the funnel plot will be performed for the assessment of publication bias? Statistical tests already exists. Also, do the authors plan to do something is publication bias is present? Once again, I also have a confusion about the outcomes investigated (lack of consistency between abstract, objectives and statistical part).

6.1. Which software will be used for analyses?

Response:

All statistical analyses will be conducted utilizing Review Manager 5.1. 

We added the following sentence in the manuscript to clarify this.

Lines 326-327

All statistical analyses will be conducted utilizing Review Manager 5.1.

6.2. Why only a visual inspection of the funnel plot will be performed for the assessment of publication bias? do the authors plan to do something is publication bias is present?

Response:

A visual inspection of the funnel plot is commonly used as a quick and simple way to check for publication bias in systematic reviews and meta-analyses. However, we understand that visual inspection is only a preliminary screening tool and may not accurately detect all forms of publication bias. Therefore, we will consider adding post-hoc analyses, such as trim and fill method or the Egger's linear regression test, to estimate the effects of the missing studies and correct for publication bias.

We modified and added the following sentence in the manuscript to clarify this.

Lines 323-326

Finally, the potential publication bias will be assessed through visual inspection of the funnel plot and/or Trim and fill method. To address publication bias, we may employ trim and fill or Egger's linear regression test in post-hoc analyses.

6.3. Once again, I also have a confusion about the outcomes investigated (lack of consistency between abstract, objectives and statistical part)

Response:

As we mentioned above, primary outcome will be quality of life, patient-reported health status, and all adverse events related to IMT. We have made the descriptions of each section more consistent.

Reviewer #1: 

The recommendations for study protocol of the systematic review: The efficacy of inspiratory muscle training in patients with coronary artery disease: Protocol for a systematic review and meta-analysis, PONE-D-22-16997

1. I believe that first, the authors can expose in the pilot search strategy which studies in the main databases answer the research question to understand the effect of IMT on anginal symptoms in coronary artery disease. In the introduction, on line 62; refers to just a single RCT demonstrating the effect of IMT in improving vascular endothelial function and oxide bioavailability in normal adults. Perhaps a more comprehensive search strategy can provide more data to justify that there are studies in the literature that support the proposed systematic review.

There is a RCT performed by Huzmel et al, showing the effect of IMT on CAD patients.

This paper showed that IMT improved respiratory and peripheral muscle strength, functional exercise capacity, pulmonary functions, health-related quality of life in CAD patients with stable angina.

To provide more background data, we added the following sentences in the Introduction.

Lines 90-97

Huzmel et al demonstrated that IMT improved not only functional exercise capacity and pulmonary functions but also health-related quality of life in patients with CAD [10]. However, there is limited data on the benefits of IMT in CAD patients. To the best of our knowledge, there have been no systematic reviews conducted to assess the impact of IMT on CAD. Further research is required to investigate patient-centered outcomes in CAD patients undergoing IMT.

2. The systematic review protocol record can include in addition to the record number; the day, month and year.

Response:

According to your suggestion, we have added the following information on systematic review protocol record.

Lines 65-66

Date registered: May 10, 2022

Registration DOI: https://doi.org/10.17605/OSF.IO/GVMY7

3. Despite line 218 is reporting that the search will be comprehensive, on line 117, with information sources, is not include other databases such as; CINAHL (Cumulative Index to Nursing and allied Health, Web of Science)

Response:

We understand that the decision to include or exclude a database in a systematic review protocol is a critical step that should not be based on convenience or personal preference. It is important to consider the relevance of the database, the availability of full-text articles, the overlap with other databases, cost, time constraints, and other relevant factors. Our systematic review aims to evaluate the evidence for the effectiveness of IMT in improving health-related quality of life and patient-reported health status in patients with CAD.

In light of our research question and goals, we have carefully considered the inclusion of databases. Although CINAHL is a well-known database that is especially relevant for nursing literature, we have decided to exclude it from our search strategy. Instead, we have included PEDro, MEDLINE, and EMBASE databases, which we believe will allow us to adequately search the literature to address our research questions. In addition, we will conduct additional manual searches to ensure that we have not missed any relevant studies.

Our goal is to provide a comprehensive and objective synthesis of the existing evidence on IMT for CAD. By conducting a thorough and transparent search of the literature, we aim to minimize potential bias and retrieve the most relevant studies while avoiding irrelevant records. We hope that the results of this systematic review will contribute to the development of novel forms of training and inform future research in this area.

4. The conclusion does not fully answer the research question in question.

Response:

Since this paper is a protocol paper, no conclusions can be drawn about the research question. In Conclusion, we summarized the main objectives and methods of the systematic review and provide an overview of the expected outcomes and contributions to the field.

Lines 460-466

In conclusion, this systematic review aims to evaluate the evidence for the effectiveness of IMT in improving health-related quality of life and patient-reported health status in patients with CAD. A comprehensive search strategy and strict inclusion and exclusion criteria will be used to identify relevant studies. The results of this review will provide a synthesis of the existing evidence on IMT for CAD and inform future research in this area. It is expected that the systematic review will contribute to the development of novel training approaches and establish IMT as a potential treatment option for CAD.

Reviewer #2: 

The protocol is well written and, in my opinion, meets the criteria for publication, as this is an option of PLOS ONE.

Response:

We would like to thank Reviewer 2 for the careful review.

Finally, we would like to express our heartfelt gratitude to the editor and reviewers again.

Your sophisticated review and comments have considerably improved our manuscript, and we would like to thank you very much for the time you took to review our manuscript. 

Sincerely yours,

Yoshito Kadoya, MD, PhD

---

## [Editor Report · Decision Letter 1]

16 May 2023

PONE-D-22-16997R1The efficacy of inspiratory muscle training in patients with coronary artery disease: Protocol for a systematic review and meta-analysisPLOS ONE

Dear Dr. Chow,

Thank you for submitting your manuscript to PLOS ONE. After careful consideration, we feel that it has merit but does not fully meet PLOS ONE’s publication criteria as it currently stands. Therefore, we invite you to submit a revised version of the manuscript that addresses the points raised during the review process. Please see my comment here below.

We look forward to receiving your revised manuscript.

Kind regards,

Charlotte Beaudart

Academic Editor

PLOS ONE

Journal Requirements:

Additional Editor Comments:

Editor's comment:

I am afraid that the authors have not adequately responded to my comment on publication bias.

I suggest the authors rephrase. Publication bias should be assessed by both visual inspection of funnel plots and Eggers regression asymmetry test.

Then, IF publication bias is suspected, the authors should perform additional analyses to assess the potential impact of this publication bias on the effect size. One of the analyses can be the trim and fill method.

I invite authors to take some training on publication bias and correct the protocol accordingly.
---

## [Editor Report · Decision Letter 2]

17 Jul 2023

The efficacy of inspiratory muscle training in patients with coronary artery disease: Protocol for a systematic review and meta-analysis

PONE-D-22-16997R2

Dear Dr. Chow,

We’re pleased to inform you that your manuscript has been judged scientifically suitable for publication and will be formally accepted for publication once it meets all outstanding technical requirements.

Kind regards,

Charlotte Beaudart

Academic Editor

PLOS ONE
---

## [Editor Report · Acceptance letter]

29 Aug 2023

PONE-D-22-16997R2 

The efficacy of inspiratory muscle training in patients with coronary artery disease: Protocol for a systematic review and meta-analysis 

Dear Dr. Chow:

I'm pleased to inform you that your manuscript has been deemed suitable for publication in PLOS ONE. Congratulations! Your manuscript is now with our production department. 

Kind regards, 

on behalf of

Dr. Charlotte Beaudart 

Academic Editor

PLOS ONE